# Gestational Diabetes Mellitus in Pregnant Women with Beta-Thalassemia Minor: A Matched Case-Control Study

**DOI:** 10.3390/jcm11072050

**Published:** 2022-04-06

**Authors:** Veronica Falcone, Florian Heinzl, Bianca Karla Itariu, Theresa Reischer, Stephanie Springer, Dana Anaïs Muin, Petra Pateisky, Philipp Foessleitner, Johannes Ott, Alex Farr, Klara Rosta

**Affiliations:** 1Department of Obstetrics and Gynecology, Division of Obstetrics and Feto-Maternal Medicine, Medical University of Vienna, 1090 Vienna, Austria; veronica.falcone@meduniwien.ac.at (V.F.); florian.heinzl@meduniwien.ac.at (F.H.); theresa.reischer@meduniwien.ac.at (T.R.); stephanie.springer@meduniwien.ac.at (S.S.); dana.muin@meduniwien.ac.at (D.A.M.); petra.pateisky@meduniwien.ac.at (P.P.); philipp.foessleitner@meduniwien.ac.at (P.F.); johannes.ott@meduniwien.ac.at (J.O.); alex.farr@meduniwien.ac.at (A.F.); 2Department of Internal Medicine III, Division of Endocrinology and Metabolism Diseases, Medical University of Vienna, 1090 Vienna, Austria; bianca.itariu@meduniwien.ac.at

**Keywords:** beta-thalassemia, diabetes, gestational, pregnancy, high-risk, pregnancy complications, anemia, hypochromic

## Abstract

Pregnancy in women with thalassemia minor is considered safe. However, a higher incidence of maternal and neonatal complications in women with the disorder has been reported in the literature. This study aimed to determine whether there is an increased risk of gestational diabetes mellitus (GDM) in pregnant women with beta-thalassemia minor. We conducted a retrospective matched case-control study of 230 pregnant women who delivered at the Department of Obstetrics and Feto-Maternal Medicine at the Medical University of Vienna between the years 2008 and 2020, whereof 115 women had beta-thalassemia minor. We found no significant difference in the occurrence of GDM between the case group and control group of age and BMI-matched healthy women. However, we observed a significantly lower hemoglobin (Hb) and hematocrit (Ht) level during the first, the second, and the third trimesters of pregnancy, and postpartum (all: *p* < 0.001) among women with beta-thalassemia minor compared to the healthy controls. Neonates of women with beta-thalassemia were more likely to experience post-natal jaundice and excessive weight loss (*p* < 0.001). We conclude that GDM is not more likely to occur in pregnant women with beta-thalassemia minor. However, clinicians should be made aware of the risk of adverse maternal and neonatal outcomes. Furthermore, women with beta-thalassemia minor should undergo regular laboratory screening and multidisciplinary pregnancy care.

## 1. Introduction

Beta-thalassemia is one of the most common autosomal-recessive disorders. It is caused by a reduced (beta^+^) or absent (beta^0^) synthesis of beta globin chains of the hemoglobin (HbA1) tetramer, leading to impaired erythropoiesis [1]. Pathogenic variants of the *HBB* gene, which codes for the beta globin chains, determine either reduced production or the complete absence of beta globin chains, thus leading to varying clinical conditions [1]. While the beta-thalassemia carrier state (beta-thalassemia minor) is often asymptomatic, thalassemia intermedia comprises a heterogeneous group of thalassemia-like diseases, resulting in a wide range of microcytic anemia, from mild to severe forms.

Thalassemia major, the most severe form of thalassemia, is characterized by transfusion-dependent anemia, as well as hepatosplenomegaly, bone deformities, and multiple endocrine complications, such as hypopituitary hypogonadism and hypothyroidism [2]. Furthermore, the need of frequent blood transfusions leads to iron accumulation, which can cause potentially dangerous cardiac, hepatic, or pancreatic complications [3]. Chronic iron-mediated oxidative stress can ultimately lead to pancreatic dysfunction, causing an impaired glucose tolerance (IGT) and an increased risk of diabetes mellitus (DM) [4,5,6,7,8]. 

Moreover, peripheral insulin resistance may lead to decreased tissue glucose uptake and impaired liver glucose homeostasis [5]. The implications of thalassemia (both major and minor) on glycemic variability have been extensively studied in non-pregnant individuals [8,9]. 

To date, there are still no sufficient clinical data on pregnant women with the same disorder, as well as on the neonatal outcomes for these women. It has been suggested in the literature that there is no association between beta-thalassemia minor and poor maternal and/or neonatal outcomes. Research on the possible implications for the offspring is sparse. Several observational studies conducted on pregnant women with beta-thalassemia minor and beta-thalassemia intermedia concluded that pregnancy outcome was not associated with poor maternal and/or neonatal outcome, yet a multidisciplinary approach is still recommended [10,11,12]. Even so, for pregnant women with thalassemia intermedia, transfusion-requiring anemia has been described [10]. Further, the risk of oligohydramnios, intrauterine growth restriction, preterm labor, and C-section should be considered [13,14]. It has been postulated that pregnant women with thalassemia could be predisposed to gestational diabetes by increased insulin resistance in the liver and skeletal muscles, as well as by oxidative stress secondary to hepatic inflammation arising from increased iron turnover due to the hemolysis of microcytic erythrocytes in pregnancy [13]. Lao concluded that the presence of beta-thalassemia should encourage clinicians to screen for gestational diabetes, particularly among groups with an a priori high prevalence [13]. Changes in insulin sensitivity and secretion during pregnancy are widely described in the literature [15,16]. Insulin sensitivity decreases by up to 60% in the second half of pregnancy and can aggravate metabolic disorders caused by hereditary or acquired factors [17]. 

The present study was conducted to evaluate the risk of gestational diabetes in pregnant women affected by beta-thalassemia minor. Additionally, the course of pregnancy and the offspring’s associated outcomes were evaluated.

## 2. Materials and Methods

### 2.1. Data Acquisition

To conduct a retrospective matched case-control study, we analyzed data retrieved from the obstetrical and perinatal database at our department. Pregnant women with known thalassemia minor, who routinely presented between January 2008 and December 2020, were considered eligible for the study groups. Healthy women with singleton pregnancies without chronic conditions (e.g., thalassemia, preexisting diabetes mellitus, rheumatic diseases, HIV or hepatitis B or C infection, malignant tumor) were assigned to the control group and matched in a 1:1 ratio according to maternal age, parity, and body mass index (BMI). Data about family and medical history were also collected. For the analyses, eligible cases were systematically identified using the search terms “thalassemia” and “thalassemia minor”, followed by exclusion of cases with other forms of thalassemia, resulting in a total number of 118 cases. Three patients within the cases group were found to be suffering from preexistent diabetes mellitus: two patients had type 2 diabetes mellitus, and one patient had type 1 diabetes mellitus. These patients were excluded from the final analysis, meaning that a total of 115 patients per group were finally considered. Thalassemia was defined as microcytic hypochromic anemia confirmed by hemoglobin electrophoresis [18].

### 2.2. Outcome Measures

The occurrence of GDM over the course of pregnancy was considered the main outcome parameter. According to common guidelines, GDM was defined as an abnormal 75 g OGTT performed between 24 + 0 and 27 + 6 weeks of gestation [19,20]. Perinatal complications included preterm labor, high-grade perineal tears, post-partum hemorrhage, gestational diabetes mellitus, and preeclampsia. Additionally, the following parameters were studied: macrosomia was defined as neonatal birth weight ≥4000 g or birth weight ≥90th percentile. Preterm delivery was defined as deliveries prior to 37 + 0 weeks of gestation. A birth weight ≤2500 g or birth weight ≤10th percentile at term was defined as small for gestational age. The variable “neonatal short-complication” was defined as a composite outcome from any of the following neonatal complications: transfer to the neonatal intensive care unit (NICU), umbilical cord pH ≤7.15, neonatal acute respiratory distress syndrome (RDS), neonatal jaundice. Patients were considered anemic when hemoglobin levels were <12.0 g/dl and/or hematocrit was <35%. All relevant data were collected using the obstetric documentation software (Viewpoint^®^ Fetal Database, Version 5.6.9.17, General Electric Company, GE-Viewpoint, Munich, Germany) and patient charts. All records were anonymized prior to analysis. The study was approved by the Institutional Review Board of the Medical University of Vienna (IRB number: 1072/2020). 

### 2.3. Statistical Analyses

For statistical analysis, we used R Version 4.1.0 [21] and contributing packages (R Development Core Team, Boston, MA, USA). A total sample size of 230 patients (115 per group) resulted in a power of 80% with a two-sided level of significance of 5% and an effect size of 0.18. Categorical variables are given as numbers and percentages (%), in accordance with Pearson’s chi-squared test. Numerical variables are given as median, first quartile, and third quartile, in accordance with the Wilcoxon Rank Sum Test, while categorical variables were summarized by counts and percentage and compared using Pearson’s chi squared test. A two-sided *p* value of <0.05 was considered statistically significant.

## 3. Results

The 230 eligible pregnant women, both in the cases and control groups, were drawn from a total of 26,400 cases with delivery during the observational period. Maternal characteristics of the cases and control group are given in Table 1. There was no significant difference between the groups regarding mean age, BMI, parity, mode of conception, or mode of delivery. A total of 7 women (6.08%) in the cases group reported consanguinity with their husbands. Mean values of the standardized 75 g three-point blood glucose test at 0 (*p* = 0.26), 60 (*p* = 0.90) and 120 min (*p* = 0.80) were comparable between groups. However, women in the study group showed significantly lower hemoglobin (Hb) and hematocrit (Ht) values during the first (Hb1, Ht1), second (Hb2, Ht2), and third (Hb3, Ht3) trimester, as well as postpartum (all: *p* < 0.001), as shown on Table 1. Overall maternal complications were more common within the study group (*p* < 0.001). The sub-analysis of the complications did not show statistically significant results (glucose status: GDM treated with lifestyle modification (GDM-diet) *p* = 0.37, GDM treated pharmacologically (GDM-PT) *p* = 0.44, preeclampsia *p* = 0.66, preterm labor *p* = 0.30 post-partum hemorrhage *p* = 0.98). No statistically significant difference was found between the two groups for the delivery mode (*p* = 0.304) (Table 2).

When analyzing the perinatal outcomes, we found no significant difference between the groups regarding gestational age at delivery (*p* = 0.32), Apgar score at 5 min (*p* = 0.92), neonatal birth weight (*p* = 0.17), or arterial umbilical cord arterial blood pH at delivery (*p* = 0.75). However, neonates of women in the study group were more likely to experience complications, such as neonatal jaundice or excessive weight loss after delivery (*p* < 0.001) (Table 3).

## 4. Discussion

In the light of the profound changes in lipid and carbohydrate metabolism that occur during pregnancy [15], and considering fetal programming for metabolic diseases later in life [22,23], it is of paramount importance to define risk groups for gestational diabetes mellitus. Patients with beta-thalassemia major are at high risk of developing impaired glucose metabolism or other endocrinological disorders due to the iron overload during blood transfusions [4,5,24]. Therefore, it seems reasonable to also evaluate the risk of diabetes in thalassemia minor patients. A prospective study performed in 150 non-pregnant individuals affected by beta-thalassemia minor showed higher fasting glucose parameters within the study group. However, no significant risk for more complex metabolism disorders (e.g., metabolic syndrome) was detected [9].

Our matched case-control study revealed similar occurrence of GDM in pregnant women with beta-thalassemia minor and healthy controls. Women within the study group had a higher prevalence of anemia during and after pregnancy and a higher rate of pregnancy-associated complications compared to healthy controls. Additionally, neonates of mothers with beta-thalassemia were more likely to develop jaundice or reduced weight gain after birth.

It has been recently reported that patients who will develop GDM later in pregnancy might show higher HbA1c levels already at early gestation. However, the accuracy of this laboratory parameter in detecting pregnancy-induced glucose alterations is moderate [24]. This may be due to the prolonged half-life of red cells during iron-deficiency anemia, which can lead to false higher HbA1c levels, or to the interference of hemoglobin variants (e.g., beta-thalassemia) with HbA1c laboratory assessment [25,26]. Thus, HbA1c should not be preferred to the OGTT for GDM diagnosis, especially in patients affected by beta-thalassemia minor [26].

The analysis of other biomarkers during early pregnancy for the detection of GDM is currently an active field of research [27]. In addition to surrogate markers of glucose metabolism, pregnant women who will convert to GDM later in pregnancy show altered lipid metabolism already during the first weeks of pregnancy. In particular, the analysis of circulating sphingolipids has detected higher levels of the serum ceramide-18 (C:18-1-Cer) within GDM pregnant women, and thus, with an AUC of 0.70. Thus, the level of C:18-1-Cer could represent a potential biomarker for the early detection of GDM and should be taken into account for the development of new prediction models [28].

Consistent with the new definition of the American Diabetes Association [29], gestational diabetes was defined in our study as “diabetes, first diagnosed in the second or in the third trimester of pregnancy, that was not clearly overt diabetes prior to pregnancy”. Diagnosis was obtained by the one-step strategy, using the International Association of Diabetes and Pregnancy Study (IADPSG) cutoffs [20], and based on the results of the HAPO study [30].

We found no significant difference in GDM-diet and GDM-PT rates between women with and without beta-thalassemia minor. Owing to the retrospective character of the study, we were able to evaluate the exact results of 125/230 OGTTs, but we were not able to compare blood sugar values, as was done in previous studies [9]. However, we were able to collect information about the OGTT result (normal/abnormal) for all patients. Our results are broadly in line with Thomson’s [31] and Charoenboon’s [32] in terms of the occurrence of GDM in patients with and without beta-thalassemia minor. In contrast to Thomson et al. [31], we only analyzed patients with beta-thalassemia minor. Of note, Charoenboon et al. [32] described an incidence of pathological glucose metabolism of 7.9%, whereas within our test population, it was 19.4%. This difference might be explained by the different diagnostic tests used. Charoenboon et al. [32] screened patients for gestational diabetes with the two-step strategy, employing a 50 g 1 h glucose challenge test (GCT) followed by a 100 g 3 h glucose test and classified patients as GDM if at least two pathological values during the 3 h glucose tolerance test were detected [32]. Based on the results of Cabrera-Fernandez et al. [33], the blood sugar measurement at 180 min during the 100 g 3 h OGTT can be omitted if the Spanish Group of Diabetes and Pregnancy (GEDE) criteria are added to the prediction model. In our study, the one-step-strategy 2 h OGTT was used for GDM diagnosis. However, both strategies are accepted by the American Diabetes Association (ADA) [29].

As shown by Yumei et al. [34] and by Duran et al. [35], a dramatic increase in GDM prevalence has been registered after the introduction of the IADPSG cutoffs. Nonetheless, these researchers observed a lower rate of pregnancy-related complications, such as cesarean section, LGA (large for gestational age) fetuses, and transfer to the NICU (neonatal intensive care unit) [34]. The higher prevalence of GDM in the study group can be explained by the diagnostic test used.

Though in other studies, an increased rate of neonates with low birth weight [32,36] and placental abnormalities (e.g., calcifications, chorangiosis) [36] have been reported in women with beta-thalassemia, we found no significant difference in birth weight between the cases and control group of our study. This may be attributed to the fact that women in our study group were well cared for and intensively monitored throughout their pregnancy. Notably, we report a higher rate of perinatal complications such as neonatal transfer to the NICU or a higher rate of jaundice in our study group. Our beta-thalassemia minor patients were more likely to receive iron substitution during pregnancy and after delivery compared with the control group. A likely explanation is that thalassemia is characterized by hemolysis and impaired erythropoiesis, eventually leading to anemia [3]. Moreover, the physiological increase in plasma volume that occurs during pregnancy can cause hemodilution and exacerbate microcytic anemia typically associated with beta-thalassemia. We accept that the data presented here are slightly limited due to the retrospective character of the study, the absence of data on glycemic values, and the small sample size. Nonetheless, we conclude that pregnancy with beta-thalassemia minor may be regarded as safe [37], although our results suggest that follow-up during and after pregnancy is recommended for women who are affected by this disorder.

## 5. Conclusions

To the best of our knowledge, our study is the first to analyze the prevalence of gestational diabetes in pregnant women with beta-thalassemia minor as the main outcome. We found no significant difference regarding the main outcome between women with and without beta-thalassemia minor. However, given the elevated risk of maternal and neonatal complications in affected women, we consider it necessary to thoroughly follow up on these women within a tertiary perinatal setting. Attention should be focused on fetal biometry and maternal iron levels during antenatal care. Prospective studies are required to evaluate the long-term maternal and the neonatal effects of this disorder.

## Figures and Tables

**Table 1 jcm-11-02050-t001:** Baseline characteristics of the 230 study participants with and without beta-thalassemia minor.

Variable	*n*	Cases	*n*	Controls	*p* Value
Age	115	31 (27.5–31)	115	30 (27–30)	0.70
Parity	98	0 (0–0)	98	0 (0–0)	0.79
BMI	113	23.7 (20.8–23.7)	113	23.7 (21.3–23.7)	0.93
Contraception *ART* *Ovarian stimulation* *Spontaneous*	115	*4 (3.5%)* *0* *93 (80.8%)*	115		0.63
				*4 (3.5%* *1 (0.8%)* *108 (93.9%)*	
Hb1	39	10.3 (9.8–10.3)	38	12.85 (12.3–12.85)	<0.001
Ht1	39	34 (31–34)	38	38.15 (36.75–38.15)	<0.001
Hb2	58	9.7 (8.75–9.7)	43	12 (11.42–12)	<0.001
Ht2	58	31 (28.02–31)	43	35.25 (33–35.25)	<0.001
Hb3	54	9.7 (8.85–9.7)	43	12.3 (11.4–12.3)	<0.001
Ht3	54	30.7 (27.95–30.7)	43	35.6 (33.7–35.6)	0.001
Hbpp	82	9.5 (8.45–9.5)	115	11.3 (10.3–11.3)	<0.001
Htpp	82	27.9 (25.9–27.9)	115	31.9 (29.2–31.9)	<0.001
HbA1c	21	5.16 (4.40–6.10)	25	5.01 (4.30–6.20)	0.28
OGTT 0′	51	80 (76.75–80)	79	79.5 (73–79.5)	0.26
OGTT 60′	49	132 (123–132)	76	139 (111.5–139)	0.90
OGTT 120′	49	104 (93–104)	76	108 (89.5–108)	0.80
Consanguinity	111	7 (6.3%)	115	1 (0.8%)	0.06

Data are presented as median (first quartile–third quartile) or number (%). Abbreviations: BMI, body mass index; OGTT, oral glucose tolerance test; Hb, hemoglobin; Ht, hematocrit; Htpp, post-partum hematocrit; GDM, gestational diabetes mellitus; 1, first trimester; 2, second trimester; 3, third trimester; HbA1c, glycated hemoglobin.

**Table 2 jcm-11-02050-t002:** Maternal outcomes of the 230 study participants with and without beta-thalassemia minor.

Variable	*n*	Cases	*n*	Controls	*p* Value
Maternal complications	111	44 (39.6%)	115	17 (14.7%)	<0.001
Mode of delivery *vaginal* *instrumental* *caesarean section*	114		115		0.30
		*47 (41.2%)* *4 (3.5%)* *63 (55.2%)*		*53 (46.1%)* *8 (6.9%)* *54 (46.9%)*	
High grade perineal tears	114	3 (2.6%)	114	4 (3.5%)	0.28
Post-partum hemorrhage	114	5 (4.3%)	115	4 (3.5%)	0.98
Preterm labor	111	21 (18.9%)	115	15 (13.0%)	0.30
GDM	111	20 (18.0%)	115	24 (20.8%)	0.92
GDM-diet	111	12 (10.8%)	115	17 (14.7%)	0.69
GDM-PT	111	8 (7.2%)	115	7 (6.1%)	0.48
preeclampsia	113	2 (1.7%)	111	4 (3.6%)	0.66

Data are presented as median (first quartile–third quartile) or number (%). Abbreviations: GDM, gestational diabetes mellitus; GDM-diet, gestational diabetes mellitus with lifestyle modification; GDM-PT, gestational diabetes mellitus with pharmacotherapy.

**Table 3 jcm-11-02050-t003:** Neonatal outcomes of the 230 study participants with and without beta-thalassemia minor.

Variable	*n*	Cases	*n*	Controls	*p* Value
Gestational age at delivery	115	38 + 4 (38 + 0–38 + 4)	115	38 + 6 (38 + 0–38 + 6)	0.32
Neonatal gender *Male* *Female* *Undefined*	115		115		0.59
		*62 (53.9%)* *52 (45.2%)* *1 (0.9%)*		*57 (49.6%)* *57 (49.6%)* *1 (0.9%)*	
Apgar 5 min	115	10 (10–10)	115	10 (9–10)	0.92
Birthweight	94	3055.52 ± 672.51	98	3181.15 ± 687.80)	0.33
Birthweight percentile	94	30 (14–30)	98	38 (20–38)	0.17
Head circumference	89	34.03 ± 1.72	95	34.32 ± 2.10	0.30
Low birth weight (<2500 g)	115	15 (13.04%)	115	13 (11.3%)	0.55
Umbilical cord pH	94	7.28 (7.23–7.28)	94	7.27 (7.22–7.27)	0.75
Neonatal complications	111	28 (25.2%)	115	13 (11.3%)	<0.001

Data are presented as median (first quartile–third quartile) or number (%).

## Data Availability

The data that support the results of this study are available from the corresponding author upon reasonable request.

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
