# Peer review of "Gestational Diabetes Mellitus in Pregnant Women with Beta-Thalassemia Minor: A Matched Case-Control Study"

_jcm, 2022, doi:10.3390/jcm11072050_

Round 1

Reviewer 1 Report

Thank you for your study on the incidence of GDM in a high risk group for the development of diabetes mellitus. As you state, the identification of individuals at risk for the development of GDM has important implications on the prevention of maternal and neonatal complications during pregnancy. The overall impression is scientifically sound. In the following, you will find a few comments:

  • After reading your study, to me, it is not clear, were these cases of GDM newly diagnosed cases of GDM, or did you also include individuals with thalassemia with already known prediabetes/diabetes before pregnancy? Please specify that in the method section. 
  • In current guidelines and clinical practice, OGTT is the preferred diagnostic of GDM, as used in your study. However, I was wondering if you could include any information on HbA1c in your study. As far as I know, in thalessemia, due to anemia, the HbA1c tends do be false-low and is therefore difficult to be used in this situation, e.g. compared to individuals with type 2 diabetes mellitus. Do you have a baseline HbA1c or a HbA1c during the course of the pregnancy and could you include that in your tables (e.g. table 1)? It would be interesting to see if there is not only a difference in Hb and Ht but also a difference in HbA1c and also worth mentioning why anemia/thalassemia may influence HbA1c, but not the results from the OGTT in the discussion.
  • line 90: small typographic error, I think it should be "g" instead of "mg OGTT" there

Reviewer 2 Report

Major

  1. The authors need to mention if IRB approval was obtained, exempt or waived. 
  2. Please expand the data acquisition section to include what relevant information were extracted. It seems that family and medical history are missing. Why was this not included?
  3. Authors said sample size calculation was not done due to the retrospective nature of the study. This is not a valid reason since power analysis is based on study hypothesis and not design. It wasn't mentioned either whether the sampling method done was by convenience, quota or systematic sampling. This needs to be mentioned.
  4. The authors need to include the limitations of the study in the discussion, given the lack of sample size calculation and the nature of the study.
  5. The recommendations do not seem to reflect the findings. Since their is no added GDM risk for those with beta thalassemia minor other than adverse neonatal outcomes, routine maternal care is enough while attention should be given to neonates.

Minor

  1. Please expand the data analysis part to include tests done for continuous variables and covariates included in the model.
  2. The titles of tables 2 and 3 need to be switched.
  3. The authors may need the assistance of an English native in editing the manuscript as there are many grammar lapses in the paper starting from the abstract.
  4. References need to be updated as some dates back as far as 1986.

Reviewer 3 Report

It is a good research article. We believe that the 1:1 design is a bit short for this observational study, one case : two controls ,could easily have been carried out. Case-control designs raise their efficiency to a ratio of 1 case: 4 controls. The fact that it is a retrospective study should not circumvent the calculation of the sample size. It would let us know what alpha and beta errors we're dealing with. Two citations should be included at the end of the fourth paragraph of the discussion. "Juchnicka et al and Cabrera-Fernández et al have recently worked on this topic"

1.- Juchnicka I, Kuźmicki M, Zabielski P, Krętowski A, Błachnio-Zabielska A, Szamatowicz J. Serum C18:1-Cer as a Potential Biomarker for Early Detection of Gestational Diabetes. J Clin Med. 2022 Jan 13;11(2):384. doi: 10.3390/jcm11020384. PMID: 35054078; PMCID: PMC8781005.

2.- Cabrera Fernández S, Martín Martínez MD, De Francisco Montero C, Gabaldón Rodríguez I, Vilches Arenas Á, Ortega Calvo M.  Predictive models of gestational diabetes, a new prediction mode]. Semergen. 2021 Nov-Dec;47(8):515-520.  doi: 10.1016/j.semerg.2021.07.014. Epub 2021 Sep 9. PMID: 34509372.

The conclusions are interesting, but perhaps with a larger sample size, significance would have been reached in some of the categorical variables.

Round 2

Reviewer 2 Report

I am satisfied with the authors' revision of the work.